# Arrhythmia and Heart Rate Variability during Long Interdialytic Periods in Patients on Maintenance Hemodialysis: Prospective Observational Cohort Study

**DOI:** 10.3390/jcm12010265

**Published:** 2022-12-29

**Authors:** Ha Young Choi, Nam-Jun Cho, Samel Park, Hwamin Lee, Min Hong, Eun Young Lee, Hyo-Wook Gil

**Affiliations:** 1Department of Internal Medicine, Soonchunhyang University Cheonan Hospital, Cheonan 31151, Republic of Korea; 2Department of Medical Informatics, College of Medicine, Korea University, Seoul 02708, Republic of Korea; 3Department of Software Convergence, Soonchunhyang University, Asan 31538, Republic of Korea

**Keywords:** hemodialysis, electrocardiography, cardiac arrhythmia, wearable electronic devices

## Abstract

Sudden cardiac death among hemodialysis patients is related to the hemodialysis schedule. Mortality is highest within 12 h before and after the first hemodialysis sessions of a week. We investigated the association of arrhythmia occurrence and heart rate variability (HRV) using an electrocardiogram (ECG) monitoring patch during the long interdialytic interval in hemodialysis patients. This was a prospective observational study with 55 participants on maintenance hemodialysis for at least six months. A patch-type ECG monitoring device was applied to record arrhythmia events and HRV during 72 h of a long interdialytic period. Forty-nine participants with sufficient ECG data out of 55 participants were suitable for the analysis. The incidence of supraventricular tachycardia and ventricular tachycardia did not significantly change over time. The square root of the mean squared differences of successive NN intervals (RMSSD), the proportion of adjacent NN intervals differing by >50 ms (pNN50), and high-frequency (HF) increased during the long interdialytic interval. The gap in RMSSD, pNN50, HF, and the low-frequency/high-frequency (LF/HF) ratio between patients with and without significant arrhythmias increased significantly over time during the long interdialytic interval. The daily changes in RMSSD, pNN50, HF, and the LF/HF ratio were more prominent in patients without significant arrhythmias than in those with significant arrhythmias. The electrolyte fluctuation between post-hemodialysis and subsequent pre-hemodialysis was not considered in this study. The study results suggest that the decreased autonomic response during interdialytic periods in dialysis patients is associated with poor cardiac arrhythmia events.

## 1. Introduction

Cardiovascular events are the major cause of death among dialysis patients [1]. The incidence of sudden death in patients undergoing hemodialysis is very high [2]. Sudden cardiac death accounts for two-thirds of all cardiac deaths and one-fourth of all deaths [1,3] and the hemodialysis schedule is related to these rates [4,5,6,7]. In most countries, maintenance hemodialysis is conducted thrice weekly and consists of two 48 h breaks (short interdialytic period) and one 72 h break (long interdialytic period). Mortality is greatest during the first 12 h of starting dialysis and the last 12 h of the 72 h dialysis-free interval [4,5,6]. The role of arrhythmia patterns in this process is controversial. A recent report showed that bradycardia and asystole rather than ventricular tachycardia might be the key causes of sudden death in hemodialysis patients [8]. The report also showed that atrial fibrillation was detected in 41% of the patients [8]. In another study, non-sustained ventricular tachycardia occurred more frequently during hemodialysis or within six hours post-hemodialysis, compared to pre- or between-hemodialysis sessions [9]. They suggested that every-other-day hemodialysis preserves circadian rhythm, but a second day without dialysis is characterized by parasympathetic withdrawal.

Autonomic imbalance of the heart is associated with sudden cardiac death (SCD). Heart rate variability (HRV) is a measure of fluctuations in the autonomic system and baroreflex sensitivity. Worse HRV has been reported to be associated with the risk of atherosclerosis-related vascular complications, SCD, poor congestive heart failure outcomes, and pulmonary hypertension [10,11,12]. In hemodialysis patients, HRV predicts long-term outcomes [9,13,14]. Twenty-four or 48 h Holter electrocardiogram (ECG) recordings showed that depressed HRV was associated with increased mortality, but 24–48 h monitoring is short to study changes in HRV during the long interdialytic period. Recent technical advances support to solve the time problem [8,9]. Implantable continuous cardiac monitoring devices have facilitated the long-term monitoring of cardiac rhythm [15]. However, safety, including serious adverse events related to implantation, is one huddle to applying the device to monitor hemodialysis patients. Non-invasive, wearable, and reliable heart monitoring is an alternative tool for monitoring cardiac rhythm.

Therefore, we investigated the association between the incidence of arrhythmia and HRV using an ECG monitoring patch during the long interdialytic interval in hemodialysis patients.

## 2. Materials and Methods

### 2.1. Study Design and Participants

This prospective observational study was performed on participants with end-stage renal disease (ESRD) treated with hemodialysis in the Dialysis Unit of Soonchunhyang University Cheonan Hospital in Cheonan, Republic of Korea. The inclusion criteria were age of 19 or older and ESRD requiring HD treatment thrice weekly for at least six months. The exclusion criteria were patients with an acute illness such as infection, bleeding, or cardiovascular events for three months, a history of atrial fibrillation, a hypersensitivity reaction to an adhesive patch, or who declined to participate in the study. The study was conducted in accordance with the Declaration of Helsinki, and the protocol was approved by the Institutional Review Board (IRB) of Soonchunhyang University Cheonan Hospital (IRB number: 2020-12-034). All study participants provided written informed consent to participate.

### 2.2. Continuous ECG Monitoring by a Wearable Patch

At study enrollment, each participant was instructed on the use of the patch-type ECG monitoring device (modiCARE-MC100, SEERS Technology, Seongnam-si, Republic of Korea). The image and detailed specifications of this device are presented in Appendix A. This patch-type ECG device consists of two separate electrodes. The electrodes are round in shape and can obtain single-lead ECGs in real-time and heart rates in the 30 to 240 beats-per-minute range. The distance between the electrodes is 120 mm, and a single-lead ECG is recorded with a sampling rate of 256 Hz. The device is operated by a replaceable battery lasting about 72 h and records ECG data to a smartphone application via Bluetooth in real-time. In this study, electrodes were placed on the left chest above the standard precordial electrode location. This position complied with the optimal position for single-lead ECG devices [16]. The accuracy of this device was reported to be comparable to ambulatory ECG monitoring in the previous study [17].

After a Friday or Saturday hemodialysis session, a study coordinator applied the ECG patch to the participants. The participants were instructed to keep the patch attached and the gateway smartphone nearby during the long interdialytic period. The ECG patch was detached during a Monday or Tuesday hemodialysis session. The 72 h continuous ECG data were uploaded to the mobiCARE database (SEERS Technology, Seongnam-si, Republic of Korea) and interpreted by a cardiologist (H.Y.C.). The data recorded during the hemodialysis session were excluded from the analysis because of time inconsistency between participants.

### 2.3. Discrimination of Arrhythmias

In this study, we obtained single-lead continuous ECG data from each participant. To validate the arrhythmic burden in patients on hemodialysis, we reviewed all raw ECG data and categorized irregular heartbeats into six arrhythmia events according to ECG morphology. First, if the QRS width was more than 120 ms and occurred earlier than expected for the next sinus impulse, it was defined as a ventricular ectopy beat (VE), which is also known as a premature ventricular complex. When two VEs were observed consecutively, it was defined as a ventricular couplet. Three or more VEs and an HR of more than 100 beats-per-minute were defined as ventricular tachycardia (VT). Additionally, compared to the P-wave of normal sinus rhythm, if abnormal P-wave morphology was observed, or if the P-wave was not clearly observed and QRS complexes of normal width were seen, it was classified as a supraventricular ectopy beat (SVE), also known as a premature atrial complex. When two SVEs were observed consecutively, it was defined as a supraventricular couplet. Three or more consecutive SVEs and an HR of more than 100 beats-per-minute were classified as supraventricular tachycardia (SVT). Significant arrhythmia was defined as the composite of supraventricular tachycardia and ventricular tachycardia.

### 2.4. HRV Measurements

Before analyzing the data, the ECG recordings were preprocessed to exclude noise and other artifacts. R peaks were automatically detected by software (mobiCARE, SEERS Technology, Seongnam-si, Republic of Korea) and visually inspected by experienced researchers. Time-domain and frequency-domain parameters were generated following the recommendations of the Task Force of the European Society of Cardiology and the North American Society of Pacing and Electrophysiology [18]. The mean heart rate (HR), the standard deviation of the NN interval (SDNN), the square root of the mean squared differences of successive NN intervals (RMSSD), the SD of the average NN intervals calculated over 5 min periods of the entire recording (SDANN), and the proportion of adjacent NN intervals differing by >50 ms (pNN50) were measured for time-domain analysis. Frequency-domain analysis was performed using a Welch periodogram with a 256-s window and 6.25% overlap. The adopted parameters and frequency bands for each were very-low-frequency (0.003 to 0.04 Hz, VLF), low-frequency (0.04 to 0.15 Hz, LF), high-frequency (0.15 to 0.40 Hz, HF), and LF/HF ratio.

### 2.5. Clinical Covariates

We collected the demographic characteristics of the participants at the time of the ECG patch application from the electronic medical records. The characteristics included age, gender, post-hemodialysis weight, hemodialysis vintage, beta-blocker use, cause of ESRD, and comorbidities including diabetes, hypertension, coronary artery disease, and cerebrovascular accident. Laboratory variables, including serum albumin, potassium, total calcium, phosphorus, intact parathyroid hormone, ferritin, and transferrin saturation were also collected. These laboratory values were acquired from the last regular pre-hemodialysis blood sampling.

### 2.6. Statistical Analysis

Statistical analyses were performed using R version 4.1.3 (The R Foundation for Statistical Computing, Vienna, Austria). Categorical variables are expressed as counts (percentage), normally distributed continuous variables as means ± SD, and non-normally distributed continuous variables as medians (interquartile ranges). The percentage of arrhythmia occurrence per hour was calculated as the number of patients with a specific event divided by the total number of patients per hour. We used the number of patients in which the event occurred instead of the number of all events per hour because multiple events in some patients can significantly impact the outcomes. Pearson’s correlation coefficient analysis was used to calculate the correlation between time and the arrhythmia occurrence percentage or HRV parameter. We used linear mixed models to examine baseline differences and differences in the mean changes in HRV parameters between the patients with and without significant arrhythmias. We included a random intercept to allow for participant-level variability in HRV parameters. The multivariable models were adjusted for potential confounding variables, including age, sex, diabetes, coronary artery disease, beta-blocker use, ultrafiltration volume of the hemodialysis session, and diurnal variation. *p*-values of <0.05 were regarded as statistically significant, and two-tailed tests were performed in all hypothesis tests.

## 3. Results

### 3.1. Participants

Fifty-five patients on HD consented to participate. Subsequently, six participants were excluded because of insufficient ECG recording (<35 h). We included 49 participants, and their demographics, dialysis information, and baseline laboratory findings are shown in Table 1. Their mean age was 55.2 ± 10.3 years, with a post-hemodialysis weight of 62.5 ± 12.6 kg and a median dialysis vintage of 71.9 (interquartile range, 26.7–143.3) months.

### 3.2. Arrhythmia during the Interdialytic Period

First, we checked heart rate changes in the HD patients during the long interdialytic period. The heart rates were highest after completing dialysis, and the daily peak and average heart rates gradually decreased until the subsequent hemodialysis (Figure 1). The heart rates had diurnal variation, which was lowest at dawn and highest in the afternoon. During the entire interdialytic period, there were 583 VEs in 38 patients, 159 ventricular couplets in eight patients, and 35 VTs in four patients. There were 1425 SVEs in 48 patients, 307 supraventricular couplets in 38 patients, and 110 SVTs in 22 patients. The changes in the percentage of arrhythmia occurrence during the interdialytic period are presented in Figure 2. The percentage of SVT and VT occurrence per hour did not significantly change over time (correlation between VT and time: *R* = 0.059, *p* = 0.636; correlation between SVT and time: *R* = 0.227, *p* = 0.067). However, the percentage of VE occurrence per hour and that of ventricular couplets decreased over time (VE: *R* = −0.301, *p* = 0.014; ventricular couplet: *R* = −0.659, *p* < 0.001), and that of SVE and supraventricular couplets increased over time (SVE: *R* = 0.469, *p* < 0.001; supraventricular couplet: *R* = 0.263, *p* = 0.033).

### 3.3. Heart Rate Variability during the Interdialytic Period

Serial changes in HRV parameters during the long interdialytic period are presented in Figure 3. The time-domain HRV parameters included SDNN, SDANN, RMSSD, and pNN50 (Figure 3A–D). In univariable analyses, SDANN, RMSSD, and pNN50 gradually increased during the interdialytic period (SDANN: *R* = 0.264, *p* = 0.032; RMSSD: *R* = 0.722, *p* < 0.001; pNN50: *R* = 0.483, *p* < 0.001), but SDNN did not change over time (*R* = 0.073, *p* = 0.560). In the multivariable linear mixed models, RMSSD and pNN50 had positive slopes during the interdialytic period (RMSSD: mean changes per day = 0.216, *p* = 0.015; pNN50: mean changes per day = 0.559, *p* = 0.016), but SDNN and SDANN did not change significantly during the interdialytic periods (SDNN: mean changes per day = −0.021, *p* = 0.777; SDANN: mean changes per day = 0.036, *p* = 0.588). The frequency-domain HRV parameters included VLF, LF, HF, and the LF/HF ratio (Figure 3E–H). In the univariable analysis, VLF, LF, and HF gradually increased during the interdialytic period (VLF: *R* = 0.388, *p* = 0.001; LF: *R* = 0.272, *p* = 0.027; HF: *R* = 0.618, *p* < 0.001). In contrast, the LF/HF ratio decreased over time (*R* = −0.340, *p* = 0.005). In the multivariable analysis, HF still had statistical significance (mean changes per day = 0.107, *p* = 0.037), but VLF, LF, and the LF/HF ratio lost the significance revealed by univariable analysis (VLF: mean changes per day = 0.107, *p* = 0.195; LF: mean changes per day = 0.105, *p* = 0.071; LF/HF ratio: mean changes per day = 0.069, *p* = 0.680).

### 3.4. Temporal Trends in HRV Parameters According to the Presence of Arrhythmia

We defined significant arrhythmia as the composite of SVT and VT and categorized the participants into two groups: the group with significant arrhythmia (SA group) and the group without significant arrhythmia (IS group). The baseline characteristics according to significant arrhythmia are presented in Table 1. The SA group was older and had higher serum total calcium levels than the IS group. There was no significant difference in other baseline characteristics between the two groups. In the multivariable linear mixed models, the mean baseline RMSSD, pNN50, and HF values in the SA group were higher than those in the IS group (RMSSD: mean difference [*β*] = 0.446, *p* = 0.007; pNN50: *β* = 1.222, *p* = 0.003; HF: *β* = 0.243, *p* = 0.005). The two groups showed a significant difference in the daily RMSSD, pNN50, HF, and LF/HF ratio changes (RMSSD: mean changes difference per day [*β*] = −0.078, *p* = 0.001; pNN50: *β* = −0.237, *p* < 0.001; HF: *β* = −0.061, *p* < 0.001; LF/HF ratio: *β* = 0.144, *p* = 0.002) (Table 2). The temporal changes in RMSSD, pNN50, HF, and the LF/HF ratio in the SA group were relatively consistent; the daily changes in RMSSD, pNN50, HF, and the LF/HF ratio were more prominent in the IS group than in the SA group (RMSSD: mean changes per day in the IS group = 0.146 [0.113–0.178] versus the SA group = 0.053 [0.025–0.082]; pNN50: 0.290 [0.209–0.372] versus 0.060 [−0.017–0.137]; HF: 0.078 [0.059–0.098] versus 0.012 [−0.005–0.028]; LF/HF ratio: −0.118 [−0.177–−0.059] versus −0.015 [−0.074–0.043]) (Figure 4, Appendix A).

## 4. Discussion

In this continuous monitoring of ECG during a long interdialytic interval, we demonstrated that ventricular arrhythmias decreased and supraventricular arrhythmias increased over time. We also showed that in time-domain HRV, RMSSD and pNN50 significantly increased during the long interdialytic interval. In frequency-domain HRV, HF significantly increased during the long interdialytic interval. We also demonstrated that the gap in RMSSD, pNN50, and HF between the SA and IS groups increased significantly over time during the long interdialytic interval, which suggested that these parameters have clinical significance because low HRV is generally associated with cardiovascular events [19].

Sudden cardiac death, mainly related to arrhythmia, is responsible for almost 30% of all-cause mortality in hemodialysis patients [20,21]. Arrhythmias, including clinically significant arrhythmia, were common in these patients [22]. Cardiovascular death or arrhythmia-related hospitalization increased following the long interdialytic period in a large United States (US) cohort [5]. A previous report showed that clinically significant arrhythmia occurred in 67% of patients, and occurred in a periodic pattern during thrice-weekly dialysis sessions, with increased rates during the first dialysis session of the week and at the end of each interdialytic period, particularly the long interdialytic period [4]. A recent study suggested that bradycardia and asystole are important arrhythmias in the long-term interval, and atrial fibrillation frequently occurred (41%) and was temporally associated with the dialysis procedure, although the events clustered around the dialysis session itself rather than the interdialytic interval [4]. These findings suggest that changes, including autonomic dysfunction during the long interdialytic interval, could affect arrhythmia patterns. In our study, ventricular arrhythmias peaked following the dialysis procedure, then decreased, and supraventricular arrhythmias increased, both of which were related to changes in the autonomic nervous system.

HRV is a measure of fluctuations in the autonomic system and baroreflex sensitivity [23,24]. Some studies suggested that depressed HRV was associated with increased mortality in ESRD patients [25,26,27]. Increases in SDNN, AASDNN, and LF were reported after the long interdialytic interval. In our study, RMSSD, pNN50, and HF were increased during the long interdialytic interval. We investigated differences in HRV according to significant arrhythmia. The mean baseline RMSSD, pNN50, and HF values were higher in the SA group. However, temporal changes in HRV parameters (RMSSD, pNN50, and HF) were not observed in the SA group. Our findings suggest that HD patients with significant arrhythmias had impaired autonomic function, consistent with other studies [28,29]. During the long interdialytic interval, accumulation of water, electrolytes such as potassium, and uremic toxins, could directly influence heart function or indirectly affect autonomic function with related increases in arrhythmia. Our findings suggest that monitoring the HRV in HD patients with autonomic dysfunction could reduce SCD.

We used a noninvasive, wearable, and clinically available patch to monitor arrhythmias and HRV. Implantable or loop cardiac recorders have the advantage of facilitating the long-term monitoring of cardiac rhythm [30]. However, complications related to the implant procedure are a barrier to its clinical application to HD patients due to bleeding tendencies [8,31]. Patch-type monitoring could be a good alternative tool to monitor cardiac rhythms in ESRD patients. Recent patch-type monitoring could observe HR and HRV over at least 72 h, which could cover the long interval interdialytic time.

Our study had some limitations. First, our study excluded six patients due to insufficient ECG recordings. Although the patch-type ECG monitoring system is convenient, the quality of the data collection should be monitored. In clinical situations, the skin conditions of HD patients are poor [32], which could affect the adhesion of the smart patch. Second, this patch ECG has the disadvantage of a single lead, so it is sometimes difficult to distinguish SVE from VE. If the ECG is multi-lead, these are easily distinguished. However, if there is only a single lead, there is a possibility that SVE might be mistaken for VE in patients with bundle branch block. Third, there was a few days’ difference between the time of ECG patch application and the blood sampling timing. So, the effect of laboratory values, especially electrolytes, on arrhythmia occurrence might not be evaluated accurately. Furthermore, because only pre-hemodialysis values were used without post-hemodialysis measurements, the electrolyte fluctuation between post-hemodialysis and subsequent pre-hemodialysis was not considered in this study. Fourth, atrial fibrillation was not observed in our study. Atrial fibrillation is also an important arrhythmia, highly prevalent in HD patients [22,33], and is frequently observed during hemodialysis sessions. Our study did not cover the dialysis session, which might be related to the prevalence of arrhythmia. Additionally, dangerous arrhythmias, including VTs and SVTs, had low incidences in our study, which may have been because our study included many stable and young HD patients.

## 5. Conclusions

Significant changes were found in HRV parameters during the long interdialytic period, especially in RMSSD, pNN50, and HF. However, changes in HRV parameters over time were not prominent in patients with significant arrhythmias. These results suggest that decreased autonomic responses in dialysis patients are associated with the development of arrhythmic events.

## Figures and Tables

**Figure 1 jcm-12-00265-f001:**
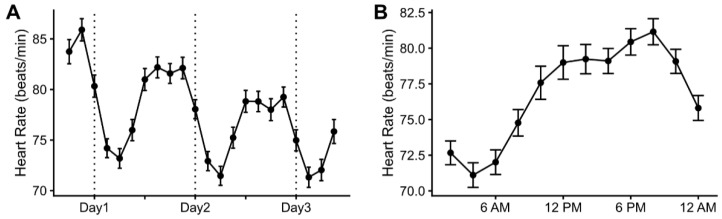
Heart rate changes in hemodialysis patients during the long interdialytic period. (**A**) Entire long interdialytic period. (**B**) Diurnal changes in heart rates. Heart rates are presented as means and standard errors in three-hour intervals.

**Figure 2 jcm-12-00265-f002:**
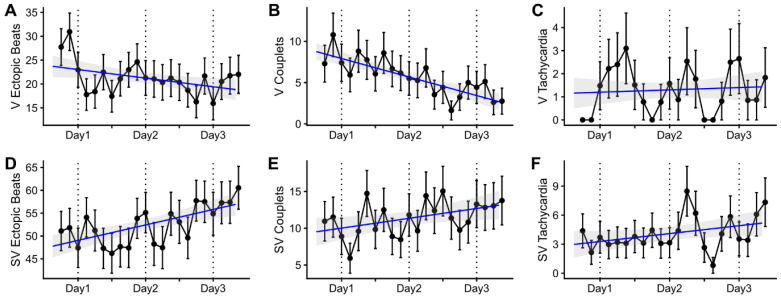
Ventricular and supraventricular arrhythmia during the long interdialytic period. Ventricular arrhythmia included ventricular ectopic beats (**A**), ventricular couplets (**B**), and ventricular tachycardia (**C**). Supraventricular arrhythmia included supraventricular ectopic beats (**D**), supraventricular couplets (**E**), and supraventricular tachycardia (**F**). The percentage of arrhythmia events was calculated as the number of patients with a specific event divided by the total number of patients per hour. The occurrence percentages are presented as means and standard errors in three-hour intervals. The blue lines and gray areas represent linear regression lines and 95% confidence intervals. V, ventricular; SV, supraventricular.

**Figure 3 jcm-12-00265-f003:**
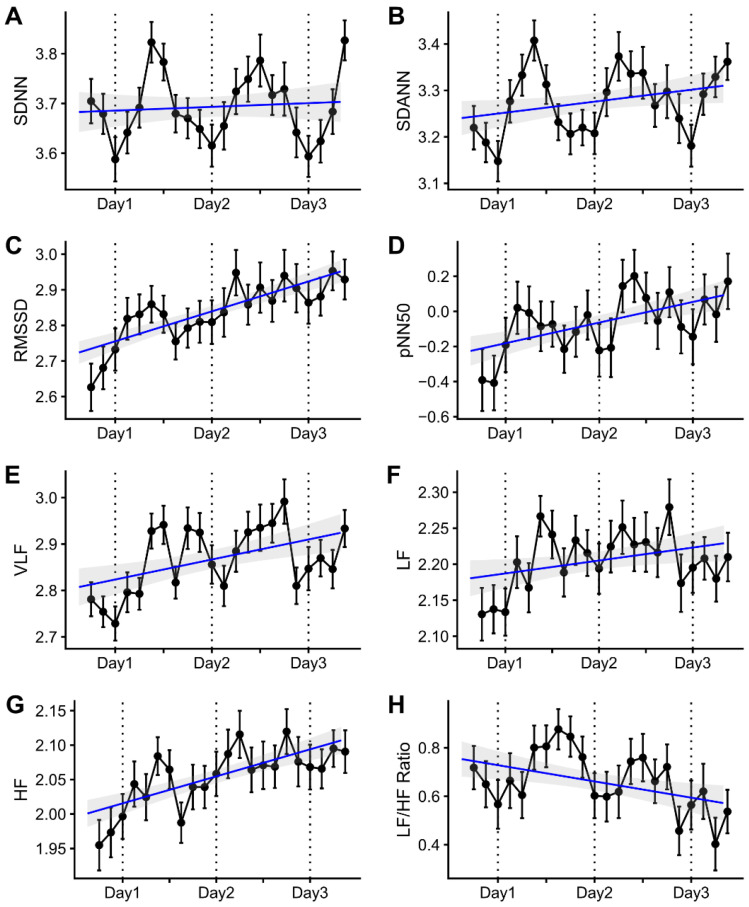
Changes in heart rate variability parameters during the long interdialytic period. The heart rate variability (HRV) parameters included time-domain parameters such as SDNN (**A**), SDANN (**B**), RMSSD (**C**), and pNN50 (**D**), and frequency-domain parameters such as VLF (**E**), LF (**F**), HF (**G**), and the LF/HF ratio (**H**). SDNN, SDANN, RMSSD, pNN50, and the LF/HF ratio values were natural log-transformed, and the VLF, LF, and HF values were square root-transformed. The HRV parameter values are presented as means and standard errors at three-hour intervals. The blue lines and gray areas represent linear regression lines and 95% confidence intervals. SDNN, standard deviation (SD) of normal-to-normal intervals during an hour; SDANN, SD of the averages of NN intervals in all five-minute segments during an hour; RMSSD, square root of the mean square of the differences between adjacent NN intervals; pNN50, percentage of the number of pairs of adjacent NN R-R intervals differing by >50 ms in the total NN intervals; VLF, very-low-frequency (0.003–0.04 Hz) power; LF, low-frequency (0.04–0.15 Hz) power; HF, high-frequency (0.15–0.40 Hz) power.

**Figure 4 jcm-12-00265-f004:**
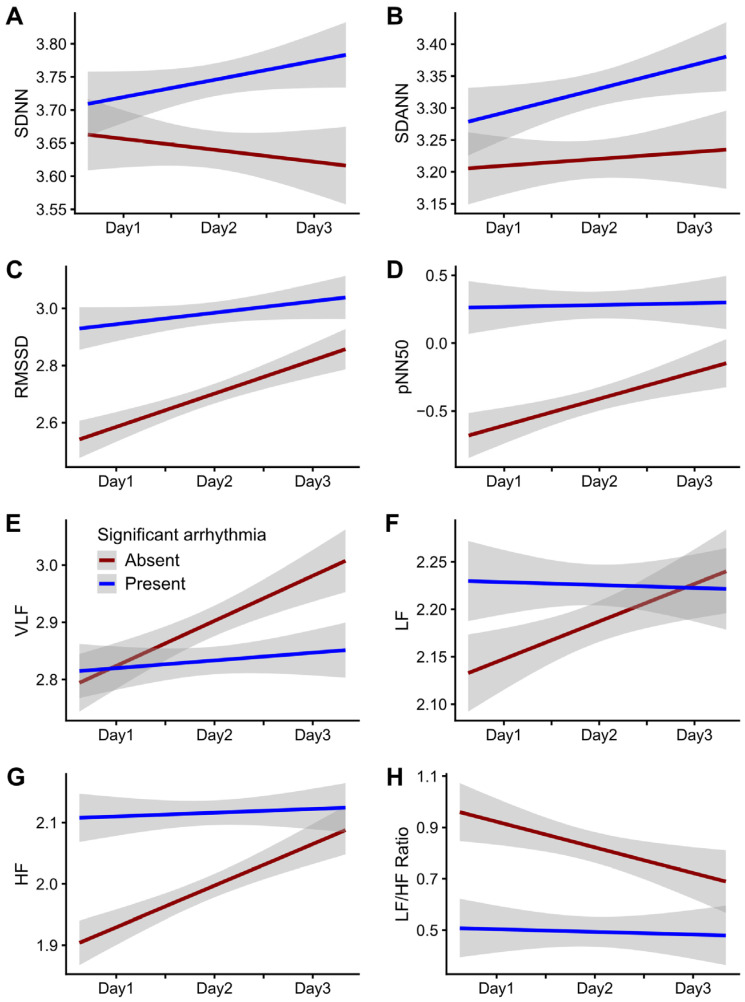
Temporal changes in HRV parameters according to significant arrhythmia. The heart rate variability (HRV) parameters included time-domain parameters such as SDNN (**A**), SDANN (**B**), RMSSD (**C**), and pNN50 (**D**), and frequency-domain parameters such as VLF (**E**), LF (**F**), HF (**G**), and the LF/HF ratio (**H**). Linear regression lines and 95% confidence intervals are presented. SDNN, SDANN, RMSSD, pNN50, and LF/HF ratio values were natural log-transformed, and the VLF, LF, and HF values were square root-transformed. SA, significant arrhythmia; SE, standard error; SDNN, standard deviation (SD) of normal-to-normal intervals during an hour; SDANN, SD of the averages of the NN intervals in all five-minute segments during an hour; RMSSD, square root of the mean square of the differences between adjacent NN intervals; pNN50, percentage of the number of pairs of adjacent NN R-R intervals differing by >50 ms in the total NN intervals; VLF, very-low-frequency (0.003–0.04 Hz) power; LF, low-frequency (0.04–0.15 Hz) power; HF, high-frequency (0.15–0.40 Hz) power.

**Table 1 jcm-12-00265-t001:** Baseline characteristics of the participants.

Characteristics	All Participants (*n* = 49)	Significant Arrhythmia		
		Absent (*n* = 26)	Present (*n* = 23)	*p*-Value
Age, years	55.2 ± 10.3	51.9 ± 9.3	59.0 ± 10.2	0.014
Male, *n* (%)	25 (51.0)	12 (46.2)	13 (56.5)	0.661
Post-hemodialysis weight, kg	62.5 ± 12.6	62.1 ± 13.5	63.0 ± 11.7	0.813
Comorbidity, *n* (%)				
Diabetes	26 (53.1)	13 (50.0)	13 (56.5)	0.865
Hypertension	48 (98.0)	26 (100.0)	22 (95.7)	0.469
Coronary artery disease	8 (16.3)	3 (11.5)	5 (21.7)	0.448
Cerebrovascular accident	1 (2.0)	1 (3.8)	0 (0.0)	1.000
Cause of ESRD, *n* (%)				0.918
Diabetic nephropathy	23 (46.9)	12 (46.2)	11 (47.8)	
Hypertensive	12 (24.5)	7 (26.9)	5 (21.7)	
Glomerulonephritis	11 (22.4)	6 (23.1)	5 (21.7)	
Other	3 (6.1)	1 (3.8)	2 (8.7)	
HD duration, months	71.9 (26.7, 143.3)	66.7 (22.9, 94.6)	89.7 (27.1, 165.3)	0.131
Beta-blocker use, *n* (%)	16 (32.7)	10 (38.5)	6 (26.1)	0.537
Hemoglobin, g/dL	10.4 ± 1.3	10.7 ± 1.3	10.2 ± 1.3	0.196
Ferritin, ng/mL	275 (199, 420)	287 (182, 427)	260 (217, 386)	0.581
Transferrin saturation, %	31.2 (26.8, 42.2)	30.3 (27.7, 40.2)	35.6 (23.6, 43.4)	0.942
Albumin, g/dL	4.10 (3.80, 4.30)	4.10 (3.82, 4.27)	4.10 (3.85, 4.25)	0.793
Potassium, mmol/L	4.58 ± 0.72	4.58 ± 0.70	4.58 ± 0.76	0.978
Total calcium, mg/dL	8.96 ± 0.65	8.75 ± 0.68	9.19 ± 0.53	0.015
Phosphorus, mg/dL	4.63 ± 1.41	4.67 ± 1.48	4.58 ± 1.36	0.833
Intact PTH, pg/mL	206 (123, 346)	219 (160, 321)	157 (111, 360)	0.302

Data are presented as mean ± SD, median (interquartile range), or count (%), as appropriate. BMI, body mass index; ESRD, end-stage renal disease; HD, hemodialysis; PTH, parathyroid hormone.

**Table 2 jcm-12-00265-t002:** Differences in HRV parameters between patients with and without significant arrhythmias.

	Unadjusted Models						Multivariable Models					
	Baseline Difference			Difference in Mean Change per Day			Baseline Difference			Difference in Mean Change per Day		
	β	SE	*p*-Value	β	SE	*p*-Value	β	SE	*p*-Value	β	SE	*p*-Value
SDNN	0.040	0.109	0.717	0.035	0.018	0.055	0.085	0.109	0.439	0.039	0.020	0.056
SDANN	0.072	0.129	0.581	0.024	0.017	0.151	0.143	0.127	0.266	0.029	0.018	0.117
RMSSD	0.465	0.158	0.005	−0.093	0.022	<0.001	0.446	0.160	0.007	−0.078	0.024	0.001
pNN50	1.159	0.394	0.005	−0.231	0.057	<0.001	1.222	0.389	0.003	−0.237	0.062	<0.001
VLF	0.049	0.079	0.541	−0.054	0.021	0.009	0.059	0.086	0.491	−0.044	0.023	0.055
LF	0.133	0.085	0.126	−0.043	0.015	0.003	0.158	0.085	0.070	−0.030	0.016	0.057
HF	0.248	0.080	0.003	−0.067	0.013	<0.001	0.243	0.082	0.005	−0.061	0.014	<0.001
LF/HF ratio	−0.487	0.223	0.033	0.103	0.042	0.015	−0.386	0.203	0.062	0.144	0.046	0.002

The statistical results were acquired from linear mixed models using Satterthwaite approximations for the degrees of freedom. The difference in HRV parameters between patients with and without significant arrhythmias was calculated, where the reference was the values of patients without significant arrhythmias (IS group). The models included time, the presence of significant arrhythmia, and their interaction. The multivariable models were adjusted for age, sex, the presence of diabetes, the presence of coronary artery disease, beta-blocker use, ultrafiltration volume of the last hemodialysis, and diurnal variation. SDNN, SDANN, RMSSD, pNN50, and the LF/HF ratio values were natural log-transformed, and the VLF, LF, and HF values were square root-transformed. SE, standard error; SDNN, standard deviation (SD) of normal-to-normal intervals during an hour; SDANN, SD of the averages of NN intervals in all five-minute segments during an hour; RMSSD, square root of the mean square of the differences between adjacent NN intervals; pNN50, percentage of the number of pairs of adjacent NN R-R intervals differing by >50 ms in the total NN intervals; VLF, very-low-frequency (0.003–0.04 Hz) power; LF, low-frequency (0.04–0.15 Hz) power; HF, high-frequency (0.15–0.40 Hz) power.

## Data Availability

The datasets used and/or analyzed during the current study are available from the corresponding author upon reasonable request.

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
