# Peer review of "Arrhythmia and Heart Rate Variability during Long Interdialytic Periods in Patients on Maintenance Hemodialysis: Prospective Observational Cohort Study"

_jcm, 2022, doi:10.3390/jcm12010265_

Round 1

Reviewer 1 Report

The manuscript by Choi et al investigated arrhythmia and heart rate variability during long interdialytic periods in a prospective cohort study. The authors conclude that the decreased autonomic response during interdialytic periods in dialysis patients is associated with poor cardiac arrhythmia events. This is a very interesting manuscript as it provides information about arrhythmia and HRV over the complete time range when the patients receive no treatment. The results are presented well and the manuscript is well written.

Here are some comments:

15: “The hemodialysis schedule is related to sudden cardiac death”: meaning of the sentence is not clear

15/16: “and mortality is the highest 15 within 6 hours after a long interdialytic period”: this is also not clear; the long interdialytic interval is 72h – to which timepoint is the 6 hours referring to?

40/41: “Mortality increases after a long interdialytic period, within six hours after the end of a hemodialysis session [5,7].”: here the same; after the following session? So the next dialysis session after the 72h break?

71: why is inclusion criterion 20 and not 18 years?

71: Only prevalent patients were included: this could be mentioned in the abstract.

72: How was clinically stable defined?

157: “excluded” should better fit in this context than “withdew”

Table 1: Cause of ESRD: Hypertensive should rather be hypertension or a word is missing

167: Figure S2 could be included in the main document as it contains nice results

Figure 3: The legend for blue and red curves should be at a central position

Author Response

Reviewer 1

15: “The hemodialysis schedule is related to sudden cardiac death”: meaning of the sentence is not clear

Response: Thank you for the good comment. We clarify this sentence as follows: “Sudden cardiac death among hemodialysis patients is related to the hemodialysis schedule.”

15/16: “and mortality is the highest 15 within 6 hours after a long interdialytic period”: this is also not clear; the long interdialytic interval is 72h – to which timepoint is the 6 hours referring to?

40/41: “Mortality increases after a long interdialytic period, within six hours after the end of a hemodialysis session [5,7].”: here the same; after the following session? So the next dialysis session after the 72h break?

Response: We reviewed the relevant studies about the timing of mortality in hemodialysis patients. Mortality was greatest during the first 12 hours of starting dialysis and the last 12 hours of the long interdialytic period. We revised the contents in the Abstract and Introduction sections.

71: why is inclusion criterion 20 and not 18 years?

Response: In Korea, the legal adult is the age of 19 or older. We rechecked the IRB documents relevant to our study and found that the age cutoff was 19. So, we modified that part, “age over 20 years” to “age of 19 or older”. There was no difference in the included participants.

71: Only prevalent patients were included: this could be mentioned in the abstract.

Response: As the reviewer pointed out, it is necessary to present that the participants were receiving hemodialysis stably. We supplemented that part of the abstract with the paragraph “for at least six months”.

72: How was clinically stable defined?

Response: We revised the ambiguous paragraph “who were not clinically stable” to more detailed term “patients with an acute illness such as infection, bleeding, or cardiovascular events for three months”.

157: “excluded” should better fit in this context than “withdew”

Response: We changed the word “withdrew” to “were excluded”.

Table 1: Cause of ESRD: Hypertensive should rather be hypertension or a word is missing

Response: We modified the word “Hypertensive” to “Hypertension” in Table 1.

167: Figure S2 could be included in the main document as it contains nice results

Response: We moved Figure S2 from the supplement document to the main document.

Figure 3: The legend for blue and red curves should be at a central position

Response: We adjusted the position of the legend to the center of Figure 3.

Reviewer 2 Report

Comments of this reviewer are: -"modiCARE-MC100, SEERS Technology, Republic of Korea": Information on accuracy and validation of this device should be provided -As the device was applied after the dialysis session and was removed before the next dialysis session the actual interval is <72h (68h or different, depending of the dialysis session's length) - "raw ECG raw" please edit - "Laboratory variables, including serum albumin, potassium, total calcium, phosphorus, intact parathyroid hormone, ferritin, and transferrin saturation were also collected" Are these laboratory data of the last month? mean of several months? Before the first dialysis of the respective interval of the study? Before the last dialysis of the respective interval of the study? Please clarify - Table S1: It would be better to incorporate it in the Manuscript - Lack of electrolyte measurements at the start (post-dialysis) and the end (predialysis) of the interval that can affect the arrhythmia incidence is a serious limitation that should be mentioned in the limitattions section and in the abstract

Author Response

"modiCARE-MC100, SEERS Technology, Republic of Korea": Information on accuracy and validation of this device should be provided.

Response: The previous report showed that this device could be an acceptable alternative for ambulatory ECG monitoring in patients with general arrhythmia (Sensors 2021, 21(9), 3122). We cited this report.

As the device was applied after the dialysis session and was removed before the next dialysis session the actual interval is <72h (68h or different, depending of the dialysis session's length)

Response: The ECG patch devices were detached during a Monday or Tuesday hemodialysis session, not before the session. Therefore, most participants attached ECG patches for about 70–72 hours. We removed the data during the hemodialysis session because of time inconsistency between participants. We clarified it in the Methods section.

"raw ECG raw" please edit

Response: We fixed the error in the revised manuscript.

"Laboratory variables, including serum albumin, potassium, total calcium, phosphorus, intact parathyroid hormone, ferritin, and transferrin saturation were also collected" Are these laboratory data of the last month? mean of several months? Before the first dialysis of the respective interval of the study? Before the last dialysis of the respective interval of the study? Please clarify

Response: The regular blood samplings were performed monthly in our clinic, and we used the last regular samples. So, there was a few days' difference between the actual time of ECG patch application and the blood sampling timing. We added these contents to the methods section and also mentioned them in the limitation part.

Table S1: It would be better to incorporate it in the Manuscript

Response: As reviewer recommended, we incorporated Table S1 with Table 1.

Lack of electrolyte measurements at the start (post-dialysis) and the end (predialysis) of the interval that can affect the arrhythmia incidence is a serious limitation that should be mentioned in the limitattions section and in the abstract

Response: We supplemented the contents about the lack of post-hemodialysis electrolyte measurement in the limitation part.

Round 2

Reviewer 2 Report

Lack of electrolyte measurements at the start (post-dialysis) and the end (predialysis) of the interval that can affect the arrhythmia incidence is a serious limitation that should be mentioned also in the abstract

Author Response

Lack of electrolyte measurements at the start (post-dialysis) and the end (predialysis) of the interval that can affect the arrhythmia incidence is a serious limitation that should be mentioned also in the abstract. 

Response: We supplemented the contents about the lack of post-hemodialysis electrolyte measurement in the abstract.